# Joint cortical registration of geometry and function using semi-supervised learning

**Jian Li**[1,2]                                                     JLI112@MGH.HARVARD.EDU
**Greta Tuckute**[3,4]                                                   GRETATU@MIT.EDU
**Evelina Fedorenko**[3,4,5]                                              EVELINA9@MIT.EDU
**Brian L. Edlow**[1,2]                                             BEDLOW@MGH.HARVARD.EDU
**Bruce Fischl**[1,6,7*]                                           BFISCHL@MGH.HARVARD.EDU
**Adrian V. Dalca**[1,6,*]                                                 ADALCA@MIT.EDU

[1] *A. A. Martinos Center for Biomedical Imaging, Department of Radiology, Massachusetts General Hospital and Harvard Medical School.*

[2] *Center for Neurotechnology and Neurorecovery, Department of Neurology, Massachusetts General Hospital and Harvard Medical School.*

[3] *Department of Brain and Cognitive Sciences, Massachusetts Institute of Technology.*

[4] *McGovern Institute for Brain Research, Massachusetts Institute of Technology.*

[5] *Program in Speech Hearing Bioscience and Technology, Harvard University.*

[6] *Computer Science and Artificial Intelligence Laboratory, Massachusetts Institute of Technology.*

[7] *Harvard-MIT Program in Health Sciences and Technology.*

**Editors:** Accepted for publication at MIDL 2023

## Abstract

Brain surface-based image registration, an important component of brain image analysis, establishes spatial correspondence between cortical surfaces. Existing iterative and learning-based approaches focus on accurate registration of folding patterns of the cerebral cortex, and assume that geometry predicts function and thus functional areas will also be well aligned. However, structure/functional variability of anatomically corresponding areas across subjects has been widely reported. In this work, we introduce a learning-based cortical registration framework, JOSA, which jointly aligns folding patterns and functional maps while simultaneously learning an optimal atlas. We demonstrate that JOSA can substantially improve registration performance in both anatomical and functional domains over existing methods. By employing a semi-supervised training strategy, the proposed framework obviates the need for functional data during inference, enabling its use in broad neuroscientific domains where functional data may not be observed. The source code of JOSA will be released to the public at https://voxelmorph.net.

**Keywords:** Cortical registration, Semi-supervised Learning

## 1. Introduction

Image registration is fundamental in medical image analysis. Deformable image registration establishes spatial correspondence between a pair of images through a dense spatial transformation that maximizes a similarity measure. Approaches for both volume-based or surface-based methods have been thoroughly studied (Maintz and Viergever, 1998; Oliveira

---

* co-senior authors with equal contribution

and Tavares, 2014). Surface-based cortical registration extracts geometric features representing brain anatomical structure, and solves the registration problem on the surface.

Cortical registration strategies achieve high accuracy in aligning complex folding patterns of the human cerebral cortex (Davatzikos and Bryan, 1996; Fischl et al., 1999b), and often improve the statistical power of group functional analysis of the brain (van Atteveldt et al., 2004; Frost and Goebel, 2012). There methods are commonly driven by geometric features that describe cortical folding patterns, such as sulcal depth and curvature (Fischl et al., 1999b; Yeo et al., 2010; Conroy et al., 2013), that are often assumed to predict function. Functional regions are thus commonly aligned via anatomical registration. However, functional variability of anatomically corresponding areas within subjects has also been widely reported (Steinmetz and Seitz, 1991; Fischl et al., 2008; Frost and Goebel, 2012). Regions with different functional specializations may not be accurately aligned even when a perfect anatomical registration is achieved.

We propose a diffeomorphic cortical registration framework that jointly describes the relationship between geometry and function. We build on recent unsupervised spherical registration strategies (Balakrishnan et al., 2019; Cheng et al., 2020a) and model a joint deformation field shared by geometry and function to capture the relatively large difference between subjects. We introduce deformation fields that describe relatively small variations between geometry and function within each subject. To avoid the biases of existing anatomical templates, we also jointly estimate a population-specific atlas during training (Dalca et al., 2019). We demonstrate this method via a semi-supervised training strategy using task functional magnetic resonance imaging (fMRI) data. In contrast to the term "unsupervised" used in the literature (Balakrishnan et al., 2019; Cheng et al., 2020a), here we borrow the term "semi-supervised" to describe our strategy to highlight that auxiliary maps, such as functional data, can be included in training but are *not* required during training. We find that this strategy yields better registration accuracy in aligning both folding patterns and functional maps in comparison to existing approaches. In summary,

1. we propose a learning-based registration framework that jointly models the relationship between geometry and function, and estimates a multi-model population-specific atlas.

2. we develop a semi-supervised training strategy that uses task fMRI data to improve functional registration but without a need for task fMRI data during inference. This semi-supervised training framework can also be extended to any auxiliary data that could be helpful to guide spherical registration but is difficult to obtain during inference, such as parcellations, architectonic identity, transcriptomic information, molecular profiles.

3. we demonstrate experimentally that the proposed framework yields improved registration performance in both anatomical and functional domains.

## 2. Related work

### 2.1. Model-based spherical registration

Deformable registration has been extensively studied (Fischl et al., 1999b; Yeo et al., 2010; Sabuncu et al., 2010; Guntupalli et al., 2016; Robinson et al., 2014; Vercauteren et al., 2009; Nenning et al., 2017; Avants et al., 2008; Beg et al., 2005). Typical strategies often employ an iterative approach that seeks an optimal deformation field to warp a moving image to a fixed image. Methods usually involve optimization of a similarity measure

between two feature images, such as the mean squared error (MSE) or normalized cross correlation (NCC), while regularizing the deformation field to have some desired property, such as smoothness and/or diffeomorphism. Widely used techniques for cortical surface registration map the surface onto a unit sphere and establish correspondence between feature maps in the spherical space (Fischl et al., 1999a). Conventional approaches, such as FreeSurfer (Fischl et al., 1999b), register an individual subject to a probabilistic population atlas by minimizing the convexity MSE weighted by the inverse variance of the atlas convexity, in a maximum *a posteriori* formulation. These anatomical registration methods have been adapted to functional registration by minimizing MSE on functional connectivity computed from fMRI data (Sabuncu et al., 2010). Spatial correspondence can be also maximized in a non-diffeomorphic manner by finding local orthogonal transforms that linearly combine features around each local neighborhood (Guntupalli et al., 2016, 2018). Recent discrete optimization approaches iteratively align local features using spherical meshes from low-resolution to high-resolution (Robinson et al., 2014, 2018).

Diffeomorphic registration enables invertibility and preserves anatomical topology using an exponentiated Lie algebra, most often assuming a stationary velocity field (SVF) (Ashburner, 2007; Vercauteren et al., 2009). These strategies were extended to the sphere by regularizing the deformation using spherical thin plate spline interpolation (Yeo et al., 2010). Several methods align functional regions, for example using Laplacian eigen embeddings computed from fMRI data (Nenning et al., 2017). These methods are successful but solve an optimization problem for each image pair, resulting in a high computational cost.

## 2.2. Learning-based spherical registration

Recent learning-based registration methods can be categorized into supervised (Krebs et al., 2017; Sokooti et al., 2017; Yang et al., 2016; Cao et al., 2017) and unsupervised registration (Dalca et al., 2018a; Balakrishnan et al., 2019; Cheng et al., 2020a; Niethammer et al., 2019; Krebs et al., 2019; de Vos et al., 2019). Supervised registration predicts deformation fields minimizing the difference with a "ground truth" deformation. "Ground truth" maps are often generated using iterative methods or via simulation, which fundamentally limits the performance of supervised approaches (Sokooti et al., 2017; Yang et al., 2016; Cao et al., 2017).

Unsupervised registration methods employ a classical loss evaluating image similarity and deformation regularity, thus forming an end-to-end training pipeline (de Vos et al., 2019; Balakrishnan et al., 2019). Semi-supervised methods employed additional information, like segmentation maps, to guide registration without requiring them during inference (Balakrishnan et al., 2019). Recent methods extend this strategy to the spherical domain by parameterizing the brain surfaces in a 2D grid that accounts for distortions (Cheng et al., 2020a) or directly on the sphere using spherical kernels (Zhao et al., 2021).

Learning-based methods improved registration run time substantially during inference, while achieving superior or similar accuracy relative to iterative methods. However, these methods do not account for variations between geometry and function within a subject.

## 2.3. Atlas building

Atlas or template construction has been widely studied in classical iterative approaches (Collins et al., 1995; Fischl et al., 1999b; Desikan et al., 2006; Destrieux et al., 2010; Joshi

et al., 2022). These methods build atlases by repeatedly registering subjects to an estimated atlas, and estimating a new atlas by averaging the registered subjects (Dickie et al., 2017). Recent learning based methods facilitate faster atlas construction (Lee et al., 2022b,a), enabling more population-specific atlases, optionally conditioned on clinical attributes (Dalca et al., 2019; Cheng et al., 2020b; Dey et al., 2021; Ding and Niethammer, 2022).

## 3. Method

We propose **Jo**int **S**pherical registration and **A**tlas building (JOSA), a method for registration with simultaneous atlas construction, that models the anato-functional differences not only between subjects but also within each subject.

### 3.1. Generative model

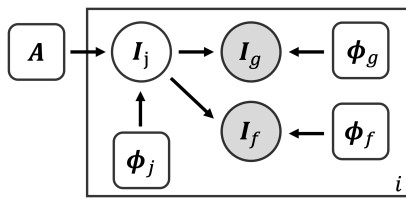

Figure 1: Graphical representation of the generative model. Circles are random variables. Rounded squares indicate parameters. Shaded quantities are observations. The big plate represents replication. $\boldsymbol{A}$ represents the global atlas, $\boldsymbol{I}$ the input image, $\boldsymbol{\phi}$ deformation field. The subscript $j$, $g$, and $f$ stand for joint, geometry, and function, respectively.

Fig. 1 shows the graphical representation for the proposed generative model. Let $\boldsymbol{A}$ be an unknown population atlas with all geometric and functional cortical features of interest. We propose a generative model that describes the formation of the subject geometric features $\boldsymbol{I}_g$ and functional features $\boldsymbol{I}_f$ by first warping the atlas $\boldsymbol{A}$ by a subject deformation field $\boldsymbol{\phi}_j$. This characterizes the differences between subjects, and results in a joint multi-feature image $\boldsymbol{I}_j$. Geometric feature $\boldsymbol{I}_g$ is formed given an additional field $\boldsymbol{\phi}_g$ that deforms the geometric features in $\boldsymbol{I}_j$, and similarly for $\boldsymbol{I}_f$ and $\boldsymbol{\phi}_f$.

**Deformation prior.** Let $\boldsymbol{\phi}_j^i, \boldsymbol{\phi}_g^i, \boldsymbol{\phi}_f^i$ be the joint, geometric, and functional deformation fields for each subject $i$, respectively. All variables in the model are subject-specific, except for the global atlas $\boldsymbol{A}$, and we omit $i$ for our derivation. We impose the deformation priors

$$\begin{aligned} p(\boldsymbol{\phi}_j) &\sim \exp\{-(\lambda_j\|\nabla \boldsymbol{u}_j\|^2 + \alpha_j\|\bar{\boldsymbol{u}}_j\|^2)\} \\ p(\boldsymbol{\phi}_g) &\sim \exp\{-\lambda_g\|\nabla \boldsymbol{u}_g\|^2\} \\ p(\boldsymbol{\phi}_f) &\sim \exp\{-\lambda_f\|\nabla \boldsymbol{u}_f\|^2\} \end{aligned} \tag{1}$$

where $\boldsymbol{u}_j$ is the spatial displacement for $\boldsymbol{\phi}_j = Id + \boldsymbol{u}_j$, $\nabla \boldsymbol{u}_j$ is its spatial gradient, and $\bar{\boldsymbol{u}}_j = 1/n \sum_i \boldsymbol{u}_j^i$, and like-wise for $\boldsymbol{u}_g$ and $\boldsymbol{u}_f$. The gradient term encourages smooth deformations, while the mean term encourages an unbiased atlas $\boldsymbol{A}$ by encouraging small average deformation over the entire dataset (Dalca et al., 2019).

**Data likelihood.** We treat the latent joint image $\boldsymbol{I}_j$ as a noisy warped atlas,

$$p(\boldsymbol{I}_j|\boldsymbol{\phi}_j; \boldsymbol{A}) = \mathcal{N}(\boldsymbol{I}_j; \boldsymbol{\phi}_j \circ \boldsymbol{A}, \sigma^2\mathbb{I}) \tag{2}$$

where $\mathcal{N}(\cdot; \boldsymbol{\mu}, \boldsymbol{\Sigma})$ is the multivariate Gaussian distribution with mean $\boldsymbol{\mu}$ and covariance $\boldsymbol{\Sigma}$, $\circ$ represents spatial transformation, $\sigma$ represents additive noise, and $\mathbb{I}$ is identity matrix. The

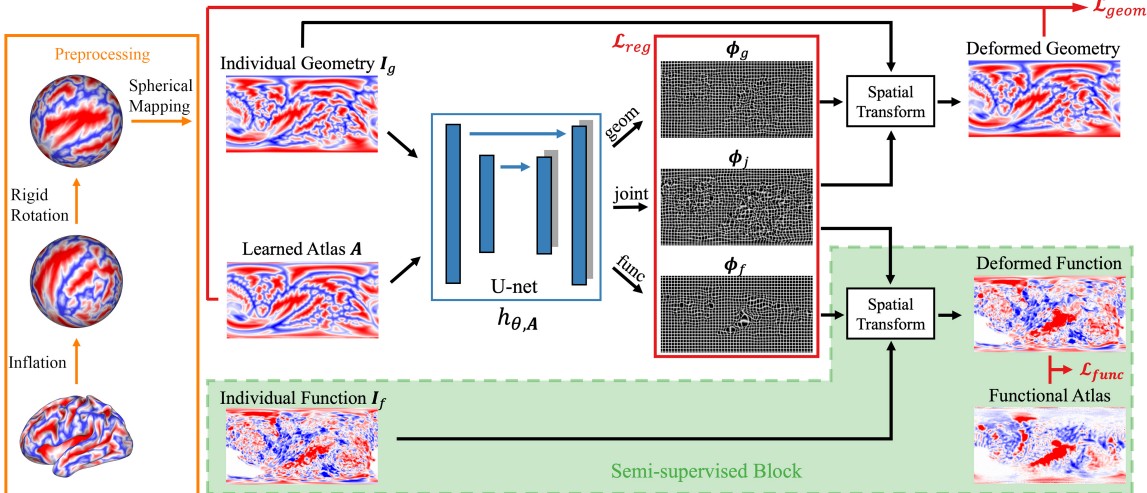

Figure 2: Network architecture and preprocessing pipeline. The network takes the geometric features from the subject and outputs one joint and two separate deformation fields for registration of folding patterns and functions separately. Task fMRI data were used for evaluating the functional loss only in a semi-supervised manner.

geometric feature image $\boldsymbol{I}_g$ is then a noisy observation of a further-moved joint image $\boldsymbol{I}_j$:

$$p(\boldsymbol{I}_g|\boldsymbol{\phi}_g, \boldsymbol{I}_j) = \mathcal{N}(\boldsymbol{I}_g; \boldsymbol{\phi}_g \circ \boldsymbol{I}_j, \sigma^2\mathbb{I}). \tag{3}$$

Therefore, the complete geometric **image likelihood** is then

$$p(\boldsymbol{I}_g|\boldsymbol{\phi}_g, \boldsymbol{\phi}_j; \boldsymbol{A}) = \int_{I_j} p(\boldsymbol{I}_g|\boldsymbol{\phi}_g, \boldsymbol{I}_j)p(\boldsymbol{I}_j|\boldsymbol{\phi}_j; \boldsymbol{A}) = \mathcal{N}(\boldsymbol{I}_g; \boldsymbol{\phi}_g \circ \boldsymbol{\phi}_j \circ \boldsymbol{A}; 2\sigma^2\mathbb{I}), \tag{4}$$

with the full derivation provided in the Appendix. We use a similar model for the functional image $\boldsymbol{I}_f$.

**Learning** Let $\boldsymbol{\Phi} = \{\boldsymbol{\phi}_j, \boldsymbol{\phi}_g, \boldsymbol{\phi}_f\}$ and $\boldsymbol{I} = \{\boldsymbol{I}_g, \boldsymbol{I}_f\}$. We estimate $\boldsymbol{\Phi}$ by minimizing the negative log likelihood,

$$
\begin{aligned}
\mathcal{L}(\boldsymbol{\Phi}|\boldsymbol{I}; \boldsymbol{A}) &= -\log p(\boldsymbol{\Phi}|\boldsymbol{I}; \boldsymbol{A}) = -\log p(\boldsymbol{I}|\boldsymbol{\Phi}; \boldsymbol{A}) - \log p(\boldsymbol{\Phi}) \\
&= -\log \prod_{k \in \{g,f\}} p(\boldsymbol{I}_k|\boldsymbol{\phi}_j, \boldsymbol{\phi}_k, \boldsymbol{A}) - \log \prod_{k \in \{j,g,f\}} p(\boldsymbol{\phi}_k) \\
&= \frac{1}{2\sigma^2}\left(\|\boldsymbol{I}_g - \boldsymbol{\phi}_g \circ \boldsymbol{\phi}_j \circ \boldsymbol{A}\|^2 + \|\boldsymbol{I}_f - \boldsymbol{\phi}_f \circ \boldsymbol{\phi}_j \circ \boldsymbol{A}\|^2\right) \qquad \textcolor{red}{\leftarrow \mathcal{L}_{geom} + \mathcal{L}_{func}} \\
&\quad + \lambda_j\|\nabla\boldsymbol{u}_j\|^2 + \lambda_g\|\nabla\boldsymbol{u}_g\|^2 + \lambda_f\|\nabla\boldsymbol{u}_f\|^2 + \alpha_j\|\bar{\boldsymbol{u}}_j\|^2 + \text{const.} \quad \textcolor{red}{\leftarrow \mathcal{L}_{reg}}
\end{aligned}
\tag{5}
$$

## 3.2. Neural network approach and semi-supervision with task fMRI data

We use a neural network to approximate the function $h_{\theta,\boldsymbol{A}}(\boldsymbol{I}) = \boldsymbol{\Phi}$, where $\theta$ are network parameters. Fig. 2 shows the proposed network architecture. To work with surface-based data, the cortical surface of each subject is inflated into a sphere and then rigidly registered to an average space using FreeSurfer (Fischl, 2012). Geometric and functional features are

parameterized onto a 2D grid using a standard conversion from Cartesian coordinates to spherical coordinates, resulting in a 2D image for each input (Cheng et al., 2020a).

The network takes such a parameterized geometric image as input and outputs *three* velocity fields, each followed by an integration layer generating the corresponding deformation field. The joint deformation $\phi_j$, which models the relatively large inter-subject variance, is shared among and composed with individual deformations $\phi_g$ and $\phi_f$. We note that the separation of $\phi_g$ from $\phi_f$ enables us to explicitly model the structural-functional differences. This is of critical importance from a neuroscientific perspective because it has been shown that some brain structure predicts their function well, and others do not (Fischl et al., 2008). The losses $\mathcal{L}_{geom}$ and $\mathcal{L}_{func}$ shown in Fig. 2 represents the geometric part and the functional part of the data fidelity terms in Eq. (5). They are evaluated in the atlas and subject space, which also helps avoid atlas drift (Aganj et al., 2017) during atlas construction. $\mathcal{L}_{reg}$ represents the regularization and centrality terms which encourages smooth deformations and an unbiased estimation of the atlas.

In this study, we learn network parameters and use task fMRI data in a *semi-supervised* manner. As shown in the green block in Fig. 2, the task fMRI data and the corresponding functional atlas are not input into the neural network. Rather they are used only for evaluating the functional terms in the loss function (5). This obviates the need for functional data during inference, as the deformation fields can be inferred only using geometric features. The proposed framework is flexible in the sense that augmentation of data modality can be easily integrated into the framework for simultaneous multi-modality registration.

### 3.3. Implementation

We implemented a Unet-like network based on the core architecture in VoxelMorph (https://voxelmorph.net) (Balakrishnan et al., 2019; Dalca et al., 2019). We used a 5-layer encoder with [128, 256, 384, 512, 640] filters and a symmetric decoder followed by 2 more convolutional layers with [64, 32] filters. Each layer involves convolution, max-pooling/up-sampling, and LeakyReLU activation. The spherical parameterization leads to denser sampling grids for regions at higher latitudes. Thus we performed prior and distortion corrections identical to that described in (Cheng et al., 2020a). In short, weights proportionally to $\sin(\theta)$, where $\theta$ is the elevation, were used to correct the distortion. (Cheng et al., 2020a) also found that varying the locations of the poles in the projection had little impact on the resulting registration. The parameterized images were standardized identically but separately for structural and functional features, where the median was subtracted for each feature image followed by a division of standard deviation.

During training, we randomly sampled the training data into mini-batches with a batch size of 8. For each batch, we augmented the data by adding Gaussian random deformations with $\sigma = 4$ with proper distortion correction at each spatial location. We also augmented the data by adding Gaussian noise with $\sigma = 1$ for geometric features and $\sigma = 6$ for functional features. We used the Adam optimizer (Kingma and Ba, 2014) with an initial learning rate of $10^{-3}$. The learning rate was scheduled to decrease linearly to $10^{-4}$ within the first 500 epochs and then reduced by a factor of 0.9 if the validation loss does not decrease after every 100 epochs. The relative weights between functional loss and geometric loss were set to 0.7:0.3 empirically. We set the regularization hyperparameter $\lambda_j$ in Eq. (5) for the joint (large) deformation to be 0.1 and $\lambda_g$ and $\lambda_f$ for the individual (small) deformations to be

0.2. The atlases, as part of the network parameters, were initialized using Gaussian random noise and automatically learned during training. We used TensorFlow (Abadi et al., 2016) with Keras front-end (Chollet, 2018) and the Neurite package (Dalca et al., 2018b), and all experiments were conducted in the Dell Workstation with dual Intel Xeon Silver 6226R CPUs and an Nvidia RTX6000 GPU. The source code of JOSA will be released to the public at https://voxelmorph.net.

## 4. Experiments

**Language task fMRI data**. We used a subset consisting of 150 subjects from a large-scale language mapping study (Lipkin et al., 2022). A reading task was used to localize the language network involving contrasting sentences and lists of nonwords strings in a standard blocked design with a counterbalanced condition order across runs. Each stimulus consisted of 12 words/nonwords, and stimuli were presented one word/nonword at a time at the rate of 450 ms per word/nonword. Each stimulus was preceded by a 100 ms blank screen and followed by a 400 ms screen showing a picture of a finger pressing a button, and a blank screen for another 100ms, for a total trial duration of 6s. Experimental blocks lasted 18s, and fixation blocks lasted 14s. Each run (consisting of 5 fixation blocks and 16 experimental blocks) lasted 358s. Subjects completed 2 runs. Subjects were instructed to read attentively and press a button whenever they saw the finger-pressing picture on the screen. Structural and functional data were collected on a 3T Siemens Trio scanner. T1-weighted images were collected in 176 sagittal slices with 1 mm isotropic resolution. Functional data (BOLD) were acquired using an EPI sequence with 4 mm thick near-axial slices, 2.1 mm × 2.1 mm in-plane resolution, TR = 2,000 ms, and TE = 30 ms. We preprocessed the data using FreeSurfer v6.0.0 as described in (Lipkin et al., 2022). The subjects' surfaces were reconstructed from the T1 images (default *recon-all* parameters) and data were analyzed on the subjects' native ("self") surface. Data were not spatially smoothed. A "sentence vs. nonword" contrast t-map was generated for each subject using first-level GLM analysis based on the blocked design. We randomly split the data into a training set with 110 subjects and a validation set with the remaining 40 subjects.

**Baselines**. We use FreeSurfer (Fischl et al., 1999b), and SphereMorph (Cheng et al., 2020a) as surface registration baseline. For FreeSurfer registration, we ran `mris_register` for each validation subject to register them to the FreeSurfer average space. For SphereMorph, we trained the network to predict a single deformation field, and used the average feature maps as the fixed atlas. At test time, we used deformation field generated based on the subject's geometric features to warp each subject's functional data to the atlas space.

**Evaluation**. Qualitatively, we computed the group mean images of both the geometric and the functional data in the validation set after registration. We then visualized them by superimposing the functional group mean map with the curvature group mean map using Freeview (Fischl, 2012). Quantitatively, we measured the registration accuracy as the correlations between the registered individual data and the group mean (Cheng et al., 2020a). Specifically, let $c_g^k = corr(\boldsymbol{I}_g^k, 1/N \sum_k \boldsymbol{I}_g^k)$ be the Pearson correlation of the resulting image $\boldsymbol{I}_g^k$ to the group mean image $1/N \sum_k \boldsymbol{I}_g^k$ for subject $k$ and geometric feature $g$. We then assess the pair-wise correlation improvement $c_{g,JOSA}^k - c_{g,JOSA}^k$ for our proposed method

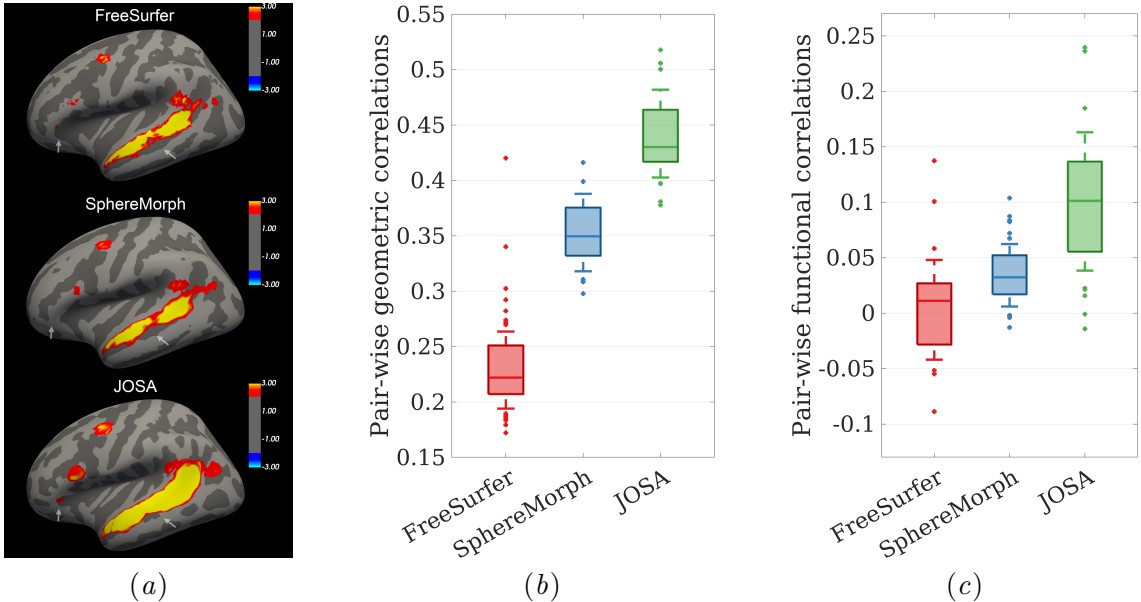

$(a)$                  $(b)$                  $(c)$

Figure 3: Comparison of registration result. (a) Average language activation map across test subjects superimposed on average curvature map. The curvature map is shown in dark gray and the thresholded language task activation map is shown using a heat map; (b) box-plot of pair-wise correlation of individual curvature map to the group mean; (c) Counterpart of (b) but for function.

JOSA, and similarly for FreeSurfer and SphereMorph (i.e., the correlation difference for the same subject after and before registration).

## 5. Results

Qualitatively, Fig. 3($a$) shows the group mean maps after registration. A better alignment leads to a cleaner group mean image with higher peaks and sharper transitions from task active to non-active regions. FreeSurfer and SphereMorph significantly improved the alignment of folding patterns but yield only marginal improvement in functional alignment. In contrast, JOSA achieved a substantially better alignment in both folding patterns and function. In particular, the predominant language region in the superior temporal gyrus shows a substantially stronger response and clearer functional boundaries, and we find additional active language-responsive regions around inferior frontal regions near Broca's area (arrows in Fig. 3($a$)), indicating an improved structural and functional alignment.

Quantitatively, Fig. 3($b$) and 3($c$) show the pair-wise registration *improvement* in correlation for each method. All three methods show substantial improvement in aligning geometry, while JOSA yielded the highest correlation increase over the rigid registration by a substantial and statistically significant margin ($p = 1.85 \times 10^{-8}$ using one-tailed Wilcoxon Signed Rank Test). Fig. 3($c$) confirmed the qualitative observation that FreeSurfer and SphereMorph marginally improved functional registration, whereas JOSA achieved a substantial and statistically significant improvement due to its separate modeling of geometry and function ($p = 2.33 \times 10^{-8}$). In addition, to illustrate the diffeomorphic property of de-

formation fields in JOSA, we computed the percentage of negative Jacobians for the testing subjects. On average, only 0.2% of the spatial locations present negative Jacobians.

Assessing computational atlases is ill-defined and often depends on their downstream utility. Fig. 4 visually compares an unlearned atlas based on FreeSurfer registration and the JOSA-learned atlas. We find that the JOSA atlas provides more anatomical definition, supporting registration with higher resolution and finer details, which may also contribute to the improved performance of the proposed method. Moreover, we conducted the ablation experiment to explore the effect of atlases. We trained two networks using identical structure as with JOSA but with the two different atlases shown in Fig. 4. Results show that the substantial improvement for geometry is mainly attributable to a better atlas, whereas the improvement in registration of function is primarily due to the separate modeling of deformation between geometry and function, as illustrated in Fig. 5 in the appendix.

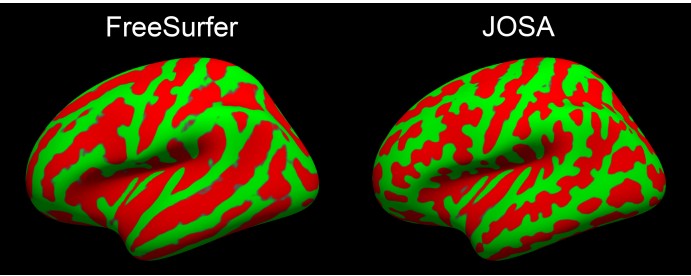

Figure 4: Atlas comparison. Curvature is shown on an inflated surface for each atlas.

## 6. Discussion

We developed JOSA, a joint geometric and functional registration framework that also estimates a population-specific atlas. JOSA yields superior performance in registering folding patterns and task-active regions in comparison to other traditional or learning-based methods. Using a semi-supervised training strategy, JOSA lifts the burden of acquiring functional data during inference, which promises to enable easier translation to scientific studies or clinical applications.

The current approach is limited by the size of the dataset as well as the single-task contrast used in the study. In particular, the hyperparameters were selected based on the validation result, which may be sub-optimal and potentially impact the model's generalizability. We plan to expand our framework to a broad range of functional data with a larger number of subjects to better explore the relationship between geometry and function. We also plan to invest our effort to more thoroughly characterize the learned atlas and analyze its contribution in spherical registration.

## Acknowledgments

Support for this research was provided in part by the BRAIN Initiative Cell Census Network grant U01MH117023, the National Institute for Biomedical Imaging and Bioengineering (R01EB023281, R01EB006758, R21EB018907, R01EB019956, P41EB030006), the National Institute on Aging (R56AG064027, R01AG064027, R01AG008122, R01AG016495, R01AG070988), the National Institute of Mental Health (UM1MH130981, R01MH123195, R01MH121885, RF1MH123195), the National Institute for Neurological Disorders and Stroke (R01NS0525851, R21NS072652, R01NS070963, R01NS083534, U01NS086625, U24NS10059103, R01NS105820, R21NS109627, RF1NS115268), the NIH Director's Office (DP2HD101400), the James S. McDonnell Foundation. This work was also made possible by the resources provided by Shared Instrumentation Grants S10RR023401, S10RR019307, and S10RR023043. Additional support was provided by the NIH Blueprint for Neuroscience Research (U01MH093765), part of the multi-institutional Human Connectome Project. Much of the computation resources required for this research was performed on computational hardware generously provided by the Massachusetts Life Sciences Center (https://www.masslifesciences.com).

In addition, BF has a financial interest in CorticoMetrics, a company whose medical pursuits focus on brain imaging and measurement technologies. BF's interests were reviewed and are managed by Massachusetts General Hospital and Partners HealthCare in accordance with their conflict of interest policies

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

## Appendix A. Derivation of Data Likelihood

The full derivation of the geometric image likelihood involves

$$p(\boldsymbol{I}_g|\boldsymbol{\phi}_g, \boldsymbol{\phi}_j; \boldsymbol{A}) = \int_{\boldsymbol{I}_j} p(\boldsymbol{I}_g|\boldsymbol{\phi}_g, \boldsymbol{I}_j)p(\boldsymbol{I}_j|\boldsymbol{\phi}_j; \boldsymbol{A}) \tag{6}$$

$$= \int_{\boldsymbol{I}_j} \mathcal{N}(\boldsymbol{I}_g; \boldsymbol{\phi}_g \circ \boldsymbol{I}_j, \sigma\mathbb{I})\,\mathcal{N}(\boldsymbol{I}_j; \boldsymbol{\phi}_j \circ \boldsymbol{A}, \sigma\mathbb{I}) \tag{7}$$

$$= \int_{\boldsymbol{I}_j} \mathcal{N}(\boldsymbol{I}_j; \boldsymbol{\phi}_g^{-1} \circ \boldsymbol{I}_g, \sigma\mathbb{I})\,\mathcal{N}(\boldsymbol{I}_j; \boldsymbol{\phi}_j \circ \boldsymbol{A}, \sigma\mathbb{I}) \tag{8}$$

$$\overset{*}{=} \int_{\boldsymbol{I}_j} \mathcal{N}(\boldsymbol{I}_j; \mu_c, \boldsymbol{\Sigma}_c)\,\mathcal{N}(\boldsymbol{\phi}_g^{-1} \circ \boldsymbol{I}_g; \boldsymbol{\phi}_j \circ \boldsymbol{A}; 2\sigma^2\mathbb{I}) \tag{9}$$

$$= \mathcal{N}(\boldsymbol{\phi}_g^{-1} \circ \boldsymbol{I}_g; \boldsymbol{\phi}_j \circ \boldsymbol{A}; 2\sigma^2\mathbb{I}) \tag{10}$$

$$= \mathcal{N}(\boldsymbol{I}_g; \boldsymbol{\phi}_g \circ \boldsymbol{\phi}_j \circ \boldsymbol{A}; 2\sigma^2\mathbb{I}) \tag{11}$$

where in $*$ we used an identity of the product of two Gaussian distributions, and $\mu_c, \Sigma_c$ are constants.

## Appendix B. Exploration of the effect of atlases

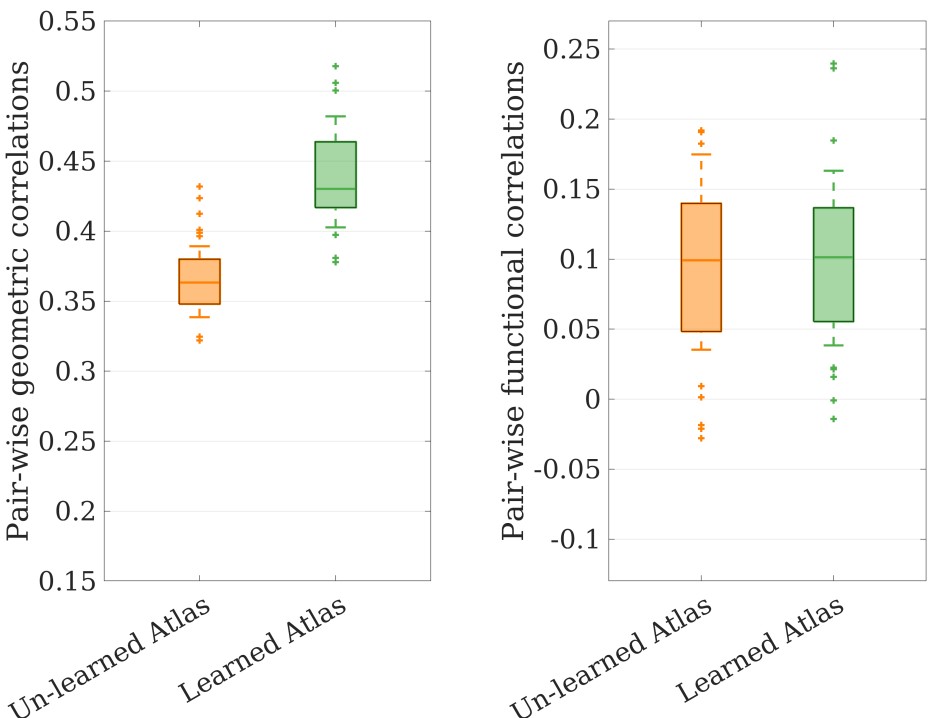

Figure 5: Box-plot of pair-wise correlation of individual curvature map to the group mean on the left and counterpart for function on the right.

