# OpenReview forum: "Joint cortical registration of geometry and function using semi-supervised learning"
_MIDL.io/2023/Conference — MIDL 2023 Poster_

### Official Review · Reviewer_5NWB · 2023-02-05

**Confidence:** 4
**Preliminary Rating:** 3
**Recommendation:** Poster

**Summary:**

This paper introduces a machine learning model for cortical registration and atlas building that draws information during the learning process from functional information, as well as the more common structural data. This is achieved by including a set of paired set of functional MRI data alongside the structural information, which is preprocessed and used as part of an additional loss function, but is never passed as an input to the model itself. The model is demonstrated on a set of language fMRI and shows an improvement in both structural and functional alignment.

**Strengths:**

This paper presents a method to integrate functional information into learning process without requiring it at inference time, and therefore reducing any burden on data acquisition. This key idea is interesting, and would likely be of interest to the community and the results indicate the method may have some benefits. The presentation quality of the work is generally good, and Figure 2 does a pretty decent job of explaining the method.

**Weaknesses:**

Although the idea is interesting, I have two major concerns about this work:
Firstly, eq. 2 contains a likelihood over the joint image $I_j$, which as described in Figure 1 is a latent variable and not observed. Therefore, how does one compute a likelihood? What is the observation used here? Moreover, it's not entirely clear what purpose $I_j$ serves? It's an partially deformed atlas I guess? This loss is not described in Figure 2 and it's not clear how this is used.
Secondly, it's never properly explained why separate transformations are used for functional and structural information, composed with a joint transformation. Moreover, as I understand it the competitor methods only use a single transformation. As such, does this not make the comparisons in the paper unfair as there are two composed SVFs that enable a more flexible transformation. For the contribution to be clearly shown, an ablation study is required to illustrate the effect of adding an additional transformation. Also, a theoretical/biological explanation for $\phi_g$ and $\phi_f$ and some illustrations of how these differ.

Minor issues:
It's not explained how the data is normalised for both functional and structural. It's mentioned that Gaussian noise $\sigma=6$ is added, is that the same for both types of data. Moreover, it states that random deformations are used $\sigma=4$ - what does this mean? What parameterisation is used?
The validation set (used for hyper-param selection) is used for testing. This is mentioned, but it weakens the results.
I'm not certain that semi-supervised learning is the best way of phrasing this - as only examples with both structural and functional information are used. I think this is one area where more clarity earlier in the paper would be helpful.
It's a little unclear what is being plotted in Figure 3 b and c - could this be stated mathematically somewhere.


**Deanonymize Review:**

no

**Paper Type:**

methodological development

**Questions To Address In The Rebuttal:**

The two major issues stated in the weaknesses need to be addressed. As it stands there are issues with the problem formulation (likelihood of $I_j$ is not clear) and with the comparison and explanation of the additional spatial transformations.

---

### Official Review · Reviewer_SJzX · 2023-02-06

**Confidence:** 3
**Preliminary Rating:** 4
**Recommendation:** Poster

**Summary:**

The authors propose a diffeomorphic model of cortical surface registration that jointly registers both structure and function.
The semisupervised approach uses functional data to simultaneously learn a global atlas that is then used to also learn joint, function and structure maps  to perform registration.

The model is trained and compared to freesurfer and an unsupervised diffeomorphic learning method, SphereMorph, demonstrating superior registration of both function and structure.

**Strengths:**

The method is very easy to understand as a solution to a problem.
The ability to learn an atlas instead of using a group average and the subsequent comparison between the learned and non-learned atlases in figure 4 is interesting. Likewise, learning a joint and a specific deformation field is a direct approach to solve the problem of registering both structure and function, and the results are convincing in showing the improvement brought on by the model.



**Weaknesses:**

The authors do not address how the projection to 2D is performed nor how or if they deal with the subsequent distortions generated. For a registration task where there is variation in shape and location of features on a sphere, this distortion will be significant.

An ablation study that uses an unlearned atlas but learns a joint distortion field would have been a valuable addition. Currently it is unclear where the improvement is coming from.

**Deanonymize Review:**

no

**Paper Type:**

methodological development

**Questions To Address In The Rebuttal:**

A sentence or two explaining how the 2d projections are made and discussing the distortion issue at least to some degree would be sufficient. A quick experiment without the learned atlas just for comparison would strengthen the paper, but I would not say that its absence merits not recommending this paper for MIDL.

---

### Official Review · Reviewer_Fsm7 · 2023-02-10

**Confidence:** 4
**Preliminary Rating:** 3
**Recommendation:** Poster

**Summary:**

In this paper, a learning-based cortical registration network is proposed with semi-supervised learning strategy to model the relationship between geometry and function of robust spherical registration. By computing the deformation prior for the joint, geometric and functional deformation field, the registration model can be optimized with a derived relationship between joint deformation field and other two fields independently.

**Strengths:**

It is interesting to model the correspondence between functional deformation field and geometric field by computing the joint deformation field in diffeomorphic setting. Figure 4 demonstrates more refine appearance than the baseline result generated from FreeSurfer.

**Weaknesses:**

This paper consists of multiple weakness that raise my concerns for acceptance:
1) From my understanding of semi-supervised learning, it consists of leveraging both labeled and unlabeled samples. In your scenario, is I_f your unlabeled samples? How do you compute I_f? From Figure 1, I only know it is computed by warping the atlas with a subject deformation field and more clarification is needed in your training scenario.

2) It will be great to further elaboration section 3.2 and I know that you are trying to minimizing the negative log likelihood. However, I cannot get the intuition behind your final derivation and seems like some innovation thoughts are behind the derivation.

3) As your proposed approach claim to be diffeomorphic, there is no derivation to demonstrate the estimation of the diffeomorphism. Also, direct inverse transform can be perform with atlas label as subject-wise prediction and generate Jacobian matrix for quantitative evaluation, which is lacking and the current result may not enough to claim with good registration.

Minor weakness:
1) More atlas template generation with deep learning have been proposed apart from brain.
Lee, Ho Hin, et al. "Multi-contrast computed tomography healthy kidney atlas." Computers in Biology and Medicine 146 (2022): 105555.
Lee, Ho Hin, et al. "Supervised deep generation of high-resolution arterial phase computed tomography kidney substructure atlas." Medical Imaging 2022: Image Processing. Vol. 12032. SPIE, 2022.

**Deanonymize Review:**

no

**Detailed Comments:**

For this paper, it is interesting to model the correspondence the cortical geometry and functionality for enhancing the registration. However, more clarification and quantitative results should be performed, in order to have high confidence to claim for good subject-wise registration with population characteristics in the generated atlas.

**Paper Type:**

methodological development

**Questions To Address In The Rebuttal:**

1) From my understanding of semi-supervised learning, it consists of leveraging both labeled and unlabeled samples. In your scenario, is I_f your unlabeled samples? How do you compute I_f?

2) What is your innovative idea behind in your section 3.2? Why do I need to optimize it like that?

3) Is it possible to perform inverse transform for both label and images to generate subject-wise label prediction for Dice evaluation, and Jacobian matrix for quantitative evaluations?

---

### Official Review · Reviewer_4KQS · 2023-02-10

**Confidence:** 3
**Preliminary Rating:** 4
**Recommendation:** Oral

**Summary:**

This paper presents JOSA, a learning-based cortical registration framework, which jointly aligns folding patterns and functional maps while simultaneously learning an optimal atlas. A semi-supervised training strategy is used in the proposed framework to obviates the need for functional data during inference.

**Strengths:**

This paper proposes a learning-based method that jointly models the relationship between geometry and function, and estimates an atlas.

A semi-supervised training strategy is proposed to improve functional registration using task fMRI data during training.

A generative model is used by incorporating the deformation prior and data likelihood.

The paper is well organized and the figures are informative.

Surface registration is an important technology for brain imaging-based analyses.

It is an interesting idea to incorporate the function atlas with the anatomical atlas.

**Weaknesses:**

Only one leaning based baseline (SphereMorph) is used in the validation.

Figure 3 shows the proposed method is more consistent with the group means. But it does not necessarily indicate a more precise method.

If the functional mapping failed due to any reason, the final registration results might worsen.





**Deanonymize Review:**

no

**Detailed Comments:**

It is not quite clear how the atlas was initialized.

It would be helpful for reproducible research if the code can be made open-source.

It is not clear why the JOSA surface is better than FreeSurfer in Fig. 4.

I am not sure if the method should be called semi-supervised learning. The functional map is still strong supervision.

It is now clear how equations (1) - (4) are directly correlated to Fig. 2.  phi_f in Fig. 2 is not clearly described in Method section.



**Paper Type:**

methodological development

**Questions To Address In The Rebuttal:**

Please see the "Weaknesses" and "Detailed Comments". I am positive about the paper, but addressing the issues and questions in the revised manuscript would make this paper with better quality and reproducibility.

---

### Meta-Review · Area_Chair_2tX3 · 2023-02-22

**Recommendation:** Accept (Poster)
**Confidence:** 5

**Metareview:**

This paper proposes a innovative approach for combined folding and functional based cortical surface alignment that explicitly accounts for the fact that folding and function is dissociated from one another across much of the surface. It makes a further novel contribution through learning at atlas which is shown to significantly improve the alignment. However, the paper could be more transparently validated against classical approaches for functional alignment and other surface learning-based alignment approaches. It is recommended to review the use of the term ‘semi-supervised’ which is not well defined.